# Batch-to-Batch Variation and Patient Heterogeneity in Thymoglobulin Binding and Specificity: One Size Does Not Fit All

**DOI:** 10.3390/jcm14020422

**Published:** 2025-01-10

**Authors:** Nicoline H. M. den Hollander, Diahann T. S. L. Jansen, Bart O. Roep

**Affiliations:** Department of Internal Medicine, Leiden University Medical Center, 2333 ZA Leiden, The Netherlands; n.h.m.den_hollander@lumc.nl (N.H.M.d.H.); d.t.s.l.jansen@lumc.nl (D.T.S.L.J.)

**Keywords:** ATG, type 1 diabetes, cytokine release syndrome, immunotherapy, autoimmune disease, precision medicine

## Abstract

**Background:** Thymoglobulin is used to prevent allograft rejection and is being explored at low doses as intervention immunotherapy in type 1 diabetes. Thymoglobulin consists of a diverse pool of rabbit antibodies directed against many different targets on human thymocytes that can also be expressed by other leukocytes. Since Thymoglobulin is generated by injecting rabbits with human thymocytes, this conceivably leads to differences between Thymoglobulin batches. **Methods:** We compared different batches for antibody composition and variation between individuals in binding to PBMC and T cell subsets, and induction of cytokines. Four different batches of Thymoglobulin were directly conjugated with Alexa-Fluor 647. Blood was collected from five healthy donors, and PBMCs were isolated and stained with Thymoglobulin followed or preceded by a panel of fluorescent antibodies to identify PBMC and T cell subsets. In addition, whole blood was incubated with unlabeled Thymoglobulin to measure cytokine induction. **Results:** Cluster analysis of flow cytometry data shows that Thymoglobulin bound to all PBMC subpopulations including regulatory T cells. However, Thymoglobulin binding was highly variable between donors and to a lesser extent between batches. Cytokines related to cytokine release syndrome were highly, but variably, increased by all Thymoglobulin batches, with strong differences between donors and moderate differences between batches. **Discussion:** The variation in Thymoglobulin binding and action between donors regarding PBMC recognition and cytokine response may underlie the different clinical responses to Thymoglobulin therapy and require personalized dose adjustment to maximize efficacy and minimize adverse side effects.

## 1. Introduction

Anti-thymocyte globulin (ATG) is used as immune suppressive induction therapy in the clinic to prevent or minimize graft rejection and is also currently being explored as intervention therapy to treat type 1 diabetes (T1D). Rabbit-ATG (Thymoglobulin) is manufactured by injecting human neonatal thymocytes into a rabbit that produces polyclonal antibodies against these cells. The polyclonal nature of Thymoglobulin is reflected in its diverse effects on the immune system. These effects include T cell depletion in blood and peripheral lymphoid tissues through complement-dependent lysis and T cell activation and apoptosis, modulation of key cell surface molecules that mediate the interactions between leukocyte and endothelium, induction of apoptosis in B cell lineages, interference with dendritic cell function and possibly the induction of regulatory T and natural killer T cells [1,2]. The diversity of anti-thymocyte antibodies, their cross-reactivity with a wide range of leukocytes and tissues, and the differences in composition of xeno-antibodies between Thymoglobulin batches may cause differences in efficacy and side effects between these therapeutic batches. Furthermore, differences in genetics and immune profile between individuals imply that Thymoglobulin may not necessarily have the same effect in everyone. Thus, such variation could lead to different immunological responses, depending on the batch and the patient that receives it. Nonetheless, Thymoglobulin therapy is now the standard of care as induction therapy in transplantation, with a great positive impact on clinical outcome and graft survival [3].

In the context of type 1 diabetes, the use of Thymoglobulin has shown mixed results. It proved indispensable in achieving insulin independence after allogenic islet transplantation into T1D recipients, compared to no induction therapy [4,5], and was superior to daclizumab in the case of pretransplant GAD65 autoantibodies to prevent rejection, but it caused CMV viremia in simultaneous kidney and pancreas transplantation in T1D [6,7]. Tapering of immune suppression therapy after long-term islet allograft failure precipitated Graves’ disease only in T1D patients that were positive for thyroid peroxidase (TPO) autoantibodies at the time of induction therapy with Thymoglobulin [8]. Thymoglobulin induction therapy also led to durable disease remission in a subgroup of newly diagnosed T1D patients treated with autologous hematopoietic stem cells [9]. However, these clinical trials also revealed that patients with high rates of CD4 or CD8 T cell autoreactivity towards islet autoantigens at the time of immunotherapy did not benefit from such procedures [10,11], indicating that Thymoglobulin may be less effective in the case of pre-existent immunity against islets. Thymoglobulin has also been assessed as a stand-alone intervention therapy at diagnosis of T1D to deplete islet autoreactive T cells, but adverse effects were found including cytokine release syndrome (CRS) with no clinical T1D improvement [12,13]. Treatment with low-dose Thymoglobulin following diagnosis of T1D preserved beta cell function [14]. This was less effective if Thymoglobulin was combined with G-CSF, with signs of accelerated disease progression in a subgroup of patients. Such variation in clinical response to treatment with Thymoglobulin as observed in T1D patients raises the question of whether this inconsistency could associate with different antibody composition or patient heterogeneity. Therefore, we investigated whether different batches of Thymoglobulin bind to different PBMC subsets in different individuals. In cases of variation in immune cell targets between donors and batches, pre-therapeutic screening of patients may prove necessary to make a deliberate decision on whether any benefits outweigh any side effects.

## 2. Materials and Methods

### 2.1. Sample Collection

Blood was collected in 9 mL sodium heparin tubes from six anonymous healthy donors following informed consent and institutional ethical approval (Leiden University Medical Center). From five donors, 200 µL of whole blood was used in the cytokine response assay and the remaining volume was used to isolate PBMCs for the Thymoglobulin target assay via Ficoll density gradient centrifugation. Preceding these assays, 200 µL whole blood from one other donor was used for a pilot experiment to ensure consistent conjugation efficiency of Thymoglobulin.

### 2.2. Thymoglobulin Conjugation Efficiency Pilot

Four different batches of Thymoglobulin (Genzyme, Cambridge, MA, USA) were obtained from left-over preparations for in-hospital-treated patients receiving transplantation. Thymoglobulin (100 µg) was conjugated with AlexaFluor-647 using the ReadyLabel™ kit (ThermoFisher, Waltham, MA, USA) following the manufacturer’s protocol. We first performed a pilot experiment where we repeated the conjugation procedure on the same Thymoglobulin batch and labeled whole blood from a single donor to assure that this procedure leads to consistent labeling efficiency. Whole blood was pre-incubated with labeled Thymoglobulin at room temperature for 30 min, lysed, washed, and incubated with either a panel of monoclonal antibodies that distinguish PBMC subsets (‘PBMC panel’) or a second panel of monoclonal antibodies distinguishing T cell subsets (‘T cell panel’). To ensure that the identification of subsets was not affected by Thymoglobulin already binding to that cell-specific marker in this staining procedure (‘Thymoglobulin first’ staining), the staining order was also reversed with panel antibodies first and Thymoglobulin second (‘Panel first’ staining), and without Thymoglobulin (‘Panel only’ staining). Flow cytometry data were obtained on a 3-Laser Aurora flow cytometer (Cytek, Fremont, CA, USA) and analyzed using FlowJo™ 10.8.1 and OMIQ© 2024 [15,16]. The mean fluorescent intensity (MFI) of AlexaFluor-647 on CD45^+^ blood cells was compared between the three labeling procedures using OMIQ© (Appendix A). This pilot experiment also highlighted the impact of erythrocyte lysis and removal required before washing and incubating with the secondary antibodies, precluding us from comparing Thymoglobulin first and Panel first staining, as the cell number (i.e., antibody concentration per cell) changes. This led us to decide to perform the rest of our studies on PBMCs instead of whole blood (Appendix A). Curiously, the background peak in the whole blood staining was absent in the PBMC staining, implying that whole blood contains cell types that are barely bound by Thymoglobulin, while all PBMCs were clearly targeted (Appendix A).

### 2.3. Thymoglobulin Target Assay

After obtaining consistent labeling results, four new batches were conjugated with label to compare PBMC and T cell labeling between different Thymoglobulin batches and between donors. The same three staining procedures (Thymoglobulin first, Panel first and Panel only) were used on 200,000 PBMCs per donor. The PBMC panel consisted of the following mouse anti-human fluorescent antibodies; anti-CD3 APC-Cy7, anti-CD45 PERCP, anti-CD8 PE-Dazzle594, anti-CD19 PE, anti-CD14 AF488, anti-CD16 PE-Cy7, anti-CD15 BV605, anti-CD56 BV510 (BioLegend, San Diego, CA, USA) and anti-CD4 BV650 (BD Biosciences, Franklin Lakes, NJ, USA). The T cell panel antibodies consisted of anti-CD3 APC-Cy7, anti-CD45RA BV785, anti-CCR7 BV421, anti-CXCR3 BV510, anti-CCR4 BV605, anti-CCR6 APC-R700, anti-FAS PE, anti-CXCR5 PE-Cy7, anti-PD1 PE-Dazzle594 (BioLegend) and anti-CD4 BV650, anti-CD25 (2A3) BB515, anti-CD25 (M-A251) BB515, anti-CD127 PerCP-Cy5.5, anti-ICOS BV711 (BD Biosciences). Flow cytometry data were again obtained on a 3-Laser Aurora flow cytometer (Cytek) and analyzed using OMIQ©. Cells were gated to remove duplicates and dead cells and to define different PBMC and T cell subsets (Appendix A). All remaining CD45^+^ cells were used as a subsample to perform PBMC clustering analysis and all CD3^+^ cells were used for T cell clustering, which was analyzed with the opt-SNE, FLOWSOM and SPADE tools. Based on unstained samples, Thymoglobulin binding was considered positive when mean fluorescent intensity (MFI) of AlexaFluor-647 > 10^4^ (Appendix A). Although labeling efficiency was consistent in the same batch, the staining intensity between batches varied. Based on this variable MFI of Thymoglobulin batches for CD45^+^ PBMCs, the color scale was normalized between batches to detect potential variability in subpopulation targets.

### 2.4. Thymoglobulin-Induced Cytokine Response

In vitro cytokine response was investigated by incubating 200 µL of blood with 3 µg Thymoglobulin (15 µg Thymoglobulin/mL blood) at 37 °C for 4 h. Supernatant was collected after centrifugation and stored at −80 °C until all samples were acquired to perform Luminex bead-based immunoassay using Bio-Plex Pro Human Cytokine 17-plex Assay kit (Bio-Rad, Hercules, CA, USA), including the following 17 cytokines: G-CSF, GM-CSF, IFN-γ, IL-1β, IL-2, IL-4, IL-5, IL-6, IL-7, IL-8, IL-10, IL-12, IL-13, IL-17, MCP-1, MIP-1β and TNF-α. The cytokine responses were analyzed with the opt-SNE and Clustered Heatmap tool in OMIQ©. Differences between donors and between batches for each cytokine were statistically tested with ANOVA (GraphPad Prism 9.5.1).

## 3. Results

### 3.1. Thymoglobulin Binding Differs Between Donors

We first tested for variability between Thymoglobulin batches and between donors in terms of binding PBMCs. Cluster analysis of flow cytometry data confirmed that Thymoglobulin bound to all PBMC and T cell subsets, whereas the staining intensity differed between donors and to a lesser extent between batches (Figure 1). Staining intensity by Thymoglobulin was relatively strong in donor 2 and weak in donor 5. Besides donor variation, we also observed some batch variation, such as higher staining intensity of batch 2 compared to batch 4 in donor 1 (Figure 1A) and batch 1 staining stronger in donor 2 compared to other donors (Figure 1B). Furthermore, the number of cells that are defined within a subset varied, such as CD8 T cells and monocytes by batch 4 in donor 3 compared to other batches in that same donor, indicating that small batch differences affected the identification of PBMC subsets (Figure 1A).

### 3.2. Monocytes and Effector Memory T Cells Are Strongly Bound by Thymoglobulin

Next, we compared different PBMC and T cell subsets for binding by Thymoglobulin by performing a spade analysis. The PBMC staining showed that all batches consistently bound strongly to monocytes in each donor, followed by CD8 T cells (Figure 2A). Other cell types and subsets were more variable between donors, such as B cells, which were stained stronger in donors 1, 3 and 4. In addition, Thymoglobulin mainly bound monocytes in donor 5, whereas many other PBMC subsets were also highly positive in donors 1 and 2. CD4 T cells were stained less intensely by Thymoglobulin compared to other subsets with the PBMC panel. Binding of Thymoglobulin to different T cell subsets as defined by the T cell panel was highly diverse; effector and memory T cells were strongly bound by Thymoglobulin, while staining of naive T cells was relatively weak (Figure 2B). Memory Tregs were more bound by Thymoglobulin in donor 1 and donor 2, and were consistently stained more strongly than naive Tregs in each donor and by all batches.

The staining intensity on the PBMC and T cell subsets did not provide insight into the antibody composition of the investigated batches, as the strongest stained cell type by a certain batch varied between donors. For example, batch 2 showed stronger staining in all PBMC subsets of donor 1—even NK cells—while batches 3 and 4 showed stronger staining in donor 3 (Figure 2A). Batch 4 bound mostly B cells and monocytes in donor 3 but primarily CD8 T cells in donor 2 (Figure 2A). Furthermore, memory Tregs were more strongly stained by batch 2 in donor 1 compared to other batches and donors (Figure 2B). These findings on the subsets show that monocytes and effector memory T cells are consistently highly bound in all donors and by all batches. However, some Thymoglobulin batches bind certain cell types more in one donor, which are less targeted in other donors—and vice versa.

### 3.3. Thymoglobulin Has PBMC- and T Cell-Specific Antibodies

To investigate the specificities within Thymoglobulin in more detail, competitive binding for cell-specific markers between panel antibodies and Thymoglobulin was assessed (Figure 3). Shifts within populations between Panel first and Thymoglobulin first confirm the presence of Thymoglobulin antibodies that are specific for a marker expressed by that subset which may or may not necessarily be present in the panel but changes the subset’s signal signature enough to cluster at a different location. We observed Thymoglobulin binding to all PBMC and T cell subsets without fully saturating the cell-specific binding spot, as the panel antibodies could still bind, indicating that Thymoglobulin mostly contained antibodies that bind either to other markers on the same cell or to a different binding site of that marker. Some subpopulations shifted more than others, such as in the monocyte (CD14^+^) cluster, implying that Thymoglobulin bound strongly to markers that are co-expressed by CD14^+^ cells (Figure 3A). Similarly, many shifts were found within the naive (CD45RA^+^/CCR7^+^) T cell cluster, which contains multiple cell surface molecules, precluding us from unambiguously defining certain specificities within Thymoglobulin; we could assign binding to a particular cell subset but not to specific markers thereof (Figure 3B). Differences between donors in PBMC composition and staining thereof with Thymoglobulin were again profound, but differences between batches were minor. Curiously, Thymoglobulin batch 1 affected the Panel first staining pattern within the naive T cell population of donor 1 but not in other donors, and this was not seen for the other Thymoglobulin batches, implying that there are both donor- and batch-related distinctions. Also, Thymoglobulin batch 4 left more staining with the monoclonal panel for a specific subset of naive T cells in donor 2, suggesting a different composition of antibody specificities in relation to naive T cells in this batch compared to other batches of Thymoglobulin. However, as this feature appeared distinct for donor 2, this observation also underscores differences in PBMC composition and/or marker expression between donors. This analysis underscores that Thymoglobulin mostly contains antibodies against additional markers than those present in our PBMC and T cell panel antibodies.

### 3.4. Thymoglobulin-Induced Cytokine Responses Vary Between Donors but Not Between Batches

To test whether commonly observed variation in clinical responses may relate to variability between Thymoglobulin batches and/or to variability between donors, we incubated whole blood from each donor with each batch and measured cytokine production in vitro. Thymoglobulin induced a profound increase in cytokines participating in cytokine release syndrome (CRS) in blood of all donors, albeit with variable strength between donors (Figure 4). Donors, rather than batches, clustered together according to a heatmap analysis (Figure 4A). Furthermore, cytokine levels in plasma without Thymoglobulin also clustered together, although the whole blood of donor 4 showed higher baseline levels of IL-8 and MCP-1. Thymoglobulin-induced cytokine levels were highly variable between donors, where donor 1 had the strongest response in all CRS-related cytokines except MCP-1, which was higher in donor 4 (Figure 4B). Donor 5 showed a relatively weak cytokine response compared to the other donors, which is in line with the distinct blood immune cell composition of donor 5, having a relatively low number of effector T cells and more naive T cells. Furthermore, some batches induced very strong responses in one donor but not in others. For example, batch 2 induced a high increase in IFN-y in donor 1, which was less prominent for other batches and donors. Two-dimensional opt-SNE analysis showed that donors clustered together, except for donor 4, whereas Thymoglobulin batches did not cluster together (Figure 4C). Moreover, no differences in overall cytokine levels were found between batches (Appendix A). Thymoglobulin induced cytokine release related to CRS in each donor, but the strength of the cytokine release was highly variable between donors. Altogether, these results are consistent with our previous studies revealing differences between donors, rather than batches, that may affect the clinical response to Thymoglobulin treatment.

## 4. Discussion

Since both composition of Thymoglobulin and the clinical response to Thymoglobulin therapy can differ between patients, we compared different Thymoglobulin batches for variation in antibody composition (i.e., cell targets) and binding to PBMCs from different donors. We found major differences in Thymoglobulin binding to various cell types between individuals, whereas batch differences were minor. Thymoglobulin induced strong in vitro CRS-related cytokine responses that were again highly variable between individuals but less variable between batches. These inter-individual differences may help us to better understand the observed differences in Thymoglobulin treatment efficacy in, for instance, T1D.

Major differences between donors were found in the binding intensity of Thymoglobulin to PBMCs and T cells. Every tested PBMC and T cell subset was bound by all batches of Thymoglobulin and in all donors, while the staining intensity varied between donors and between subsets. Thymoglobulin stained PBMC and T cell subsets with a similar intensity in some donors, whereas in other donors, staining intensity was higher for a specific subset, and this subset also differed between donors: CD8 T cells in donor 2, B cells in donor 3, monocytes in donor 5. Monocytes were consistently strongly bound, which is in line with other studies [17,18]. In vitro ATG treatment of PBMCs has been shown to induce cytotoxicity and depletion of approximately 80% of monocytes and lymphocytes. Indeed, CD8 T cells were depleted directly by binding ATG and indirectly via activated monocytes [17]. This process was dependent on the ATG-induced IFN-y response activating the JAK/STAT pathway to induce PD-L1/PD-1-mediated inhibition. Our results are consistent with these observations and other studies reporting strong induction of IFN-y by Thymoglobulin enhancing lymphocyte depletion [1,13,19]. Besides major differences between donors, subtle binding differences between Thymoglobulin batches were also noted. This is in line with a previous report assessing batch variability by investigating competition between ATG-Fresenius and different batches of Thymoglobulin for PBMC binding [18].

Inflammation and pre-existing immunity have been claimed to enhance Thymoglobulin potency [18]. The binding intensity of Thymoglobulin to PBMCs from a patient with acute graft rejection was 10-fold higher than to PBMCs from a stable transplant recipient [18]. Their observed correlation between immune response and binding intensity may explain the strong cytokine response and PBMC binding intensity that we observed in donor 1 compared to donor 5. Given that Thymoglobulin induced differential CRS-related responses between donors, and binding intensity was considered to associate with the degree of inflammation, it is conceivable that the clinical efficacy of Thymoglobulin also differs between patients depending on their immune status. Indeed, we previously demonstrated that autologous HSCT was not effective in inducing remission in T1D patients carrying high levels of circulating autoreactive CD8 and CD4 T cells, whereas low autoreactivity before Thymoglobulin therapy was associated with a good prognosis [11]. Our experiences from clinical islet transplantation into type 1 diabetic recipients showed a similar inverse relationship between the rate of islet autoimmunity before therapy and clinical outcome of Thymoglobulin induction therapy [10]. Likewise, newly diagnosed T1D patients that preserved beta cell function following low-dose Thymoglobulin treatment had a lower amount of circulating CD4 T cells at baseline compared to non-responding patients [20]. This would imply that low autoimmunity, rather than elevated immune status, improves the outcome of Thymoglobulin therapy. Furthermore, patients who had TPO autoantibodies developed Graves’ disease after discontinuation of immune suppressive therapy that followed Thymoglobulin induction therapy, suggesting that the immune system lost its capacity to suppress autoreactive T cells [8]. Moreover, the results from our clinical studies confirm that Thymoglobulin was ineffective in depleting pre-existent active autoreactive T cells, even in combination with cyclophosphamide [10,11,21]. Thus, responsiveness to Thymoglobulin therapy is negatively affected during inflammation, which might be due to high binding and depletion of immune regulatory cells that preferably should be preserved.

Interestingly, our data suggest that memory Tregs strongly bind Thymoglobulin, whereas naive Tregs seem to be spared. This may seem counterintuitive given the naive nature and composition of thymocytes against which Thymoglobulin is generated. On the other hand, some markers expressed by memory Tregs (e.g., CD69) are also present on thymocytes and activated T cells [22,23,24]. In addition, FAS is highly expressed in the CD25^+^/CD127^−^/CD45RA^−^ population, indicating that memory Tregs are mostly expressing this marker of apoptosis. Since the main purpose of thymic selection is to induce apoptosis in autoreactive lymphocytes, FAS might be highly expressed throughout the thymus, making FAS^+^ cells more susceptible to Thymoglobulin binding. The effect of Thymoglobulin on Tregs is currently under debate: several studies have proposed that Thymoglobulin could induce Tregs based on increased expression of CD25 and FOXP3, while other studies found a decrease in FOXP3 and increase in proinflammatory cytokines (e.g., IFN-y, IL-2) instead of anti-inflammatory cytokines (e.g., IL-10) [18,25,26,27,28]. With today’s knowledge, it is questionable whether these cells should be defined as Tregs, as these markers also represent activated T cells, while no evident suppressive capacity was found [28,29]. Recently, a clinical trial investigating Thymoglobulin in T1D patients claimed that regulatory T cells were relatively spared, as the disappearance of circulating effector T cells was more pronounced (approx. 40% vs. 60% depletion), which changed the balance between effector and regulatory T cells in favor of the latter [26]. The partial depletion of Tregs in that clinical trial may be explained by the preferential binding and potential depletion of memory Tregs and escape of naive Tregs that we noted in our present study. Since regulatory T cells are already impaired in suppressing effector T cells in T1D, their additional depletion by Thymoglobulin therapy could be the last straw to break immune tolerance and precipitate the onset of additional autoimmune diseases or T1D progression, as was observed in a subgroup of patients [8,20].

Thymoglobulin induced release of CRS-related cytokines in blood from each donor, but the strength of the response was highly variable between the donors. The Thymoglobulin-induced cytokine response of donor 1 was strongest for almost all cytokines. This observation may be related to the relatively high staining intensity of most PBMC subsets compared to other donors, including monocytes and B cells, as well as NK cells that are known to secrete IFN-y, TNF-α, GM-CSF, MIP-1β and IL-8 [30]. Indeed, earlier studies confirm that NK cells are a major target of Thymoglobulin and induce a very strong IFN-y response, which we earlier described as activating monocyte-induced killing of CD8 [31,32]. Thymoglobulin-induced cytokines seem to be more related to PBMC binding than T cell binding, as T cells were strongly bound in donor 2, but cytokine induction was low compared to other donors. No other donor-specific cytokine responses could be deduced from the Thymoglobulin binding assay. Although monocytes were consistently highly targeted in each donor, monocyte-related cytokines (e.g., IL-6, TNF-α, IL-8) were highly variable, indicating that this cytokine response could not easily be attributed to particular Thymoglobulin targets. This phenomenon may partly explain why in some patients Thymoglobulin can halt disease progression and in others it causes comorbidities or has no clinical effect, despite receiving the same Thymoglobulin batch and therapy regimen.

Our analyses could only provide few answers regarding Thymoglobulin’s targets, as our panel antibodies can only identify a selective number of PBMC subsets. Since Thymoglobulin is generated by injecting human donor thymocytes into rabbits (notably, we did not have access to the serum of non-immunized rabbits to conjugate for reference), we speculate that Thymoglobulin contains mostly antibodies that are directed against cell surface proteins that are less conserved between species. More specifically, MHC is one example of a highly polymorphic protein with strong inter-donor as well as inter-species variation. Therefore, some HLA alleles are conceivably more targeted by Thymoglobulin and—perhaps more importantly—this could explain why differences between donors are observed in our binding and cytokine assays as well as in the clinic. Indeed, the presence of MHC Class I targeting antibodies has also been observed by others [33]. Furthermore, our analyses were performed on PBMCs, whereas Thymoglobulin probably binds to many more cell types, considering the diversity of surface molecules on thymocytes. Previous studies have shown that Thymoglobulin binds to endothelial cells, in particular when these are activated [34]. In the context of T1D, activated islet endothelium in the insulitic lesion could enhance stress-induced antigen presentation and immune cell infiltration.

Thymoglobulin has been used for decades in the field of transplantation, whereas assessment as therapeutic intervention strategy in type 1 diabetes is quite recent and work in progress. The dose used in our in vitro study is based on the middle doses of a four-armed trial that is currently investigating the minimal effective dose to suppress autoimmunity while also limiting adverse events [35]. However, given the strong differences in cytokine response between donors, it will be difficult to determine a generalizable optimal dose that is applicable for everyone. Our study results underscore the still largely unmet need to investigate why some patients are more responsive to Thymoglobulin therapy than others to then define a personalized optimal dose accordingly.

## Figures and Tables

**Figure 1 jcm-14-00422-f001:**
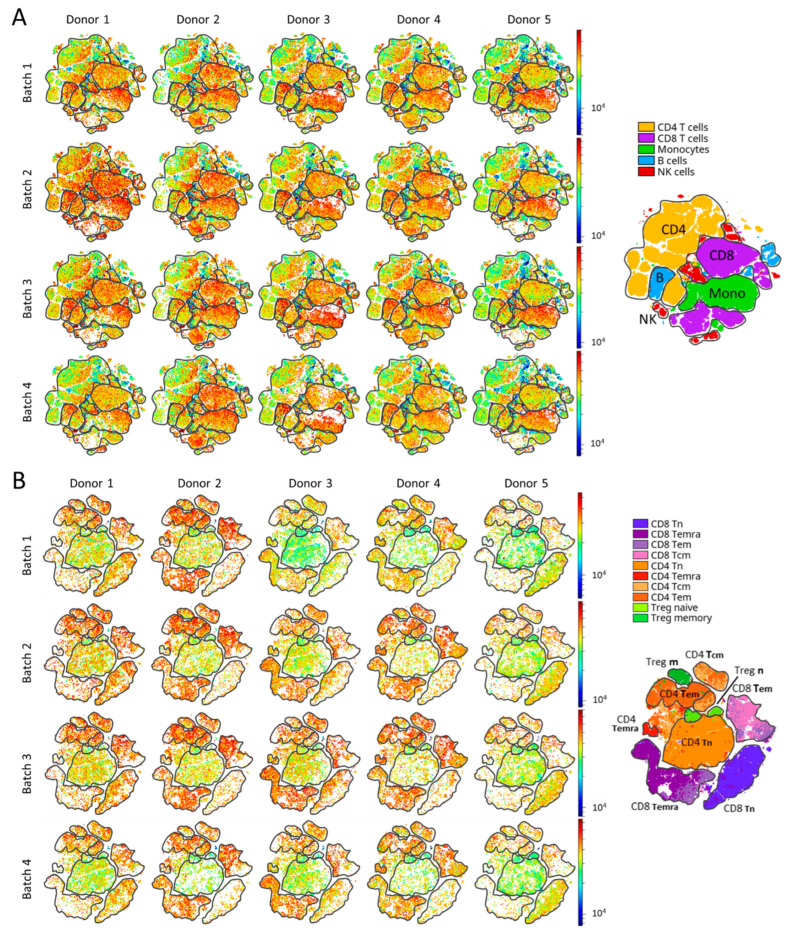
Comparison of Thymoglobulin binding to PBMC and T cell subsets between different Thymoglobulin batches and donors. Cluster analysis of flow cytometry staining with PBMC markers (**A**) and T cell markers (**B**). Mean fluorescent intensity (MFI) of Thymoglobulin per cell is shown for each donor and batch (one dot = one cell). The color range represents Thymoglobulin binding to that cell (red = more Thymoglobulin bound, blue = less Thymoglobulin bound) and the color range scale is adjusted for the fluorochrome-to-Thymoglobulin conjugation efficiency of each batch. The subsets of each panel (right map) are contoured to identify which subsets are more or less bound per batch and donor. (Mono = monocyte, B = B cell, NK = natural killer cell, CD4 = CD4^+^ T cell, CD8 = CD8^+^ T cell, Tem = effector memory, Tcm = central memory, Temra = exhausted memory, Teff = effector, Treg = regulatory T cell, m = memory, n = naive.)

**Figure 2 jcm-14-00422-f002:**
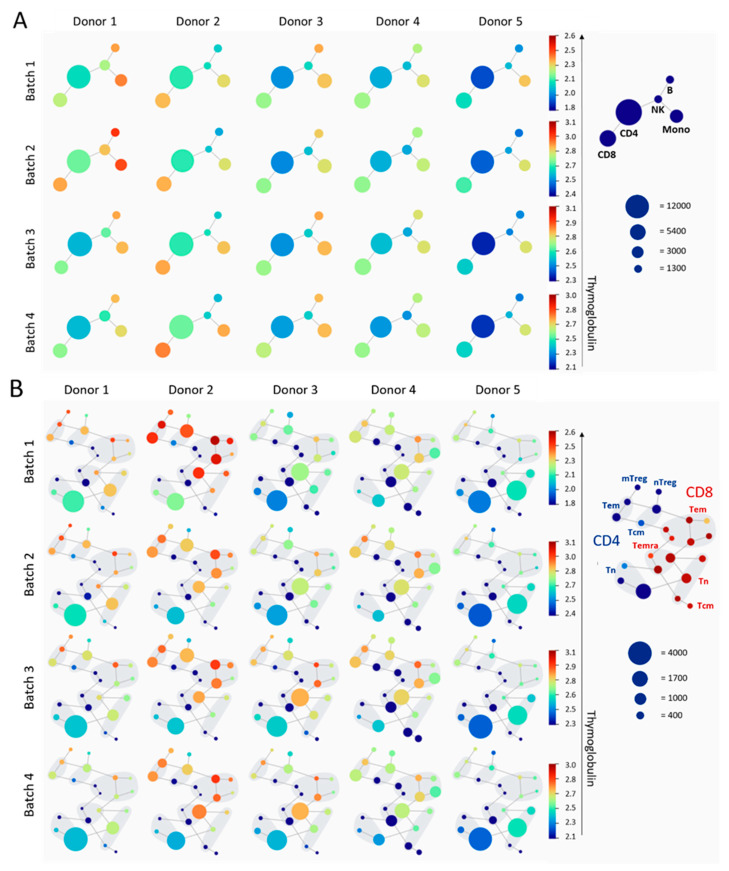
Differences in Thymoglobulin binding to PBMC and T cell subsets between Thymoglobulin batches and donors. SPADE cluster analysis of flow cytometry staining shows six clusters (dots) based on PBMC marker expression (**A**) and 22 clusters based on T cell marker expression (**B**). The MFI (mean fluorescent intensity) of Thymoglobulin is shown for each donor and batch. The size of the dot represents the number of cells per cluster. The color represents the mean binding of Thymoglobulin to all cells in that cluster and ranges from blue (less Thymoglobulin bound) to red (more Thymoglobulin bound). The color range scale is adjusted for the fluorochrome-to-Thymoglobulin conjugation efficiency of each batch. (B = B cell, NK = natural killer cell, CD8 = CD8^+^ T cell, CD4 = CD4^+^ T cell, Mono = monocyte, Tn = naive T cell, Tem = effector memory T cell, Tcm = central memory T cell, Temra = exhausted memory T cell, Treg = regulatory T cell.)

**Figure 3 jcm-14-00422-f003:**
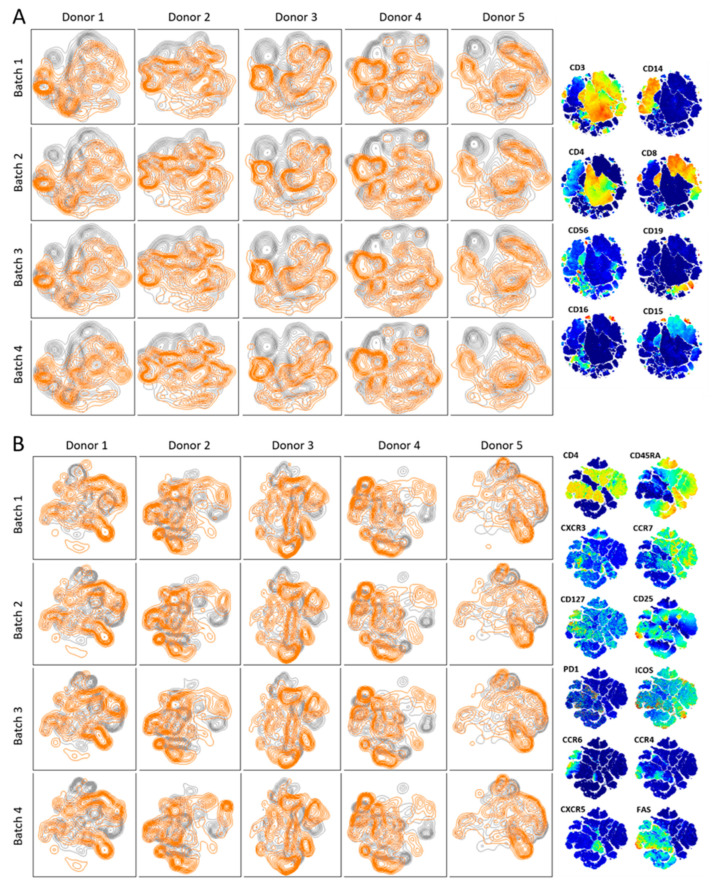
Competitive binding of Thymoglobulin to PBMC and T cell markers within different donors and batches. Cluster analysis of PBMC (**A**) and T cell (**B**) subsets from each donor and batch—with higher isobars indicating a higher cell number within a subset. Clustering is based on two different staining orders of Thymoglobulin and panel antibodies that are presented as overlays: Panel first (grey), Thymoglobulin first (orange). Distinct grey or orange isobar areas (i.e., non-overlapping) within a cluster defined by a marker indicates that Thymoglobulin contains antibodies that bind to a marker that is also expressed by that cluster, which may or may not be present in the panel. The columns on the right show the expression level per marker ranging from low (dark blue) to moderate (green) to high (orange).

**Figure 4 jcm-14-00422-f004:**
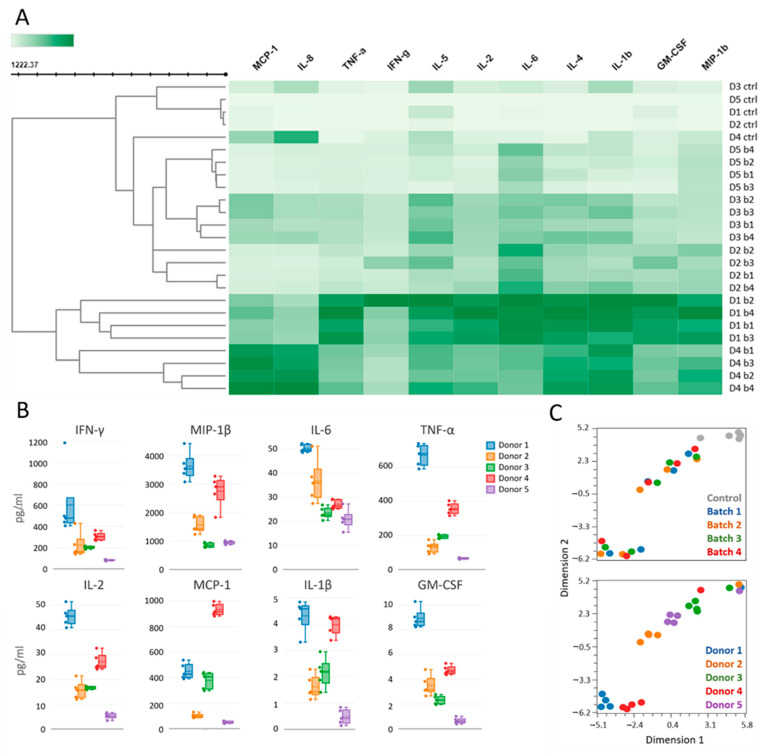
Cytokine levels after in vitro Thymoglobulin administration of different batches to whole blood from different donors. Heatmap showing cytokine levels without (control) or with different batches of Thymoglobulin (**A**). Absolute cytokine levels are presented for each donor; cytokine release syndrome (CRS)-related cytokines in upper four graphs. Cytokine levels for each cytokine were all significantly different between donors: IFN-γ (F = 5.336; *p* = 0.0071), MIP-1β (F = 39.97; *p* < 0.0001), IL-6 (F = 47.22; *p* < 0.0001), TNF-α (F = 173.6; *p* < 0.0001), IL-2 (F = 109.1; *p* < 0.0001), MCP-1 (F = 190.5; *p* < 0.0001), IL-1β (F = 45.26; *p* < 0.0001), GM-CSF (F = 100.3; *p* < 0.0001), analyzed with one-way ANOVA (**B**). Two-dimensional opt-SNE plots showing clusters of all 30 samples, with different batch colors shown in the upper graph and different donor colors shown in the lower graph (**C**). Other cytokines did not differ between donors or Thymoglobulin batches; ranges (pg/mL): G-CSF (0–43.44), IL-4 (0–2.18), IL-5 (0–31.94), IL-7 (0–3.86), IL-8 (6.03–1328.24), IL-10 (0–5.05), IL-12 (0–2.09), IL-13 (0–1.61), IL-17 (0–14.0).

## Data Availability

The raw experimental data are available upon request.

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
