# Peer review of "Batch-to-Batch Variation and Patient Heterogeneity in Thymoglobulin Binding and Specificity: One Size Does Not Fit All"

_jcm, 2025, doi:10.3390/jcm14020422_

Round 1

Reviewer 1 Report

Comments and Suggestions for Authors

The  “Batch-to-batch variation and patient heterogeneity in Thymoglobulin binding and specificity: one size does not fit all” paper describes a deep flow cytometry and induction of cytokines analysis of human Peripheral Blood Mononuclear Cells (PBMCs)  from five healthy donors and using four different batches of a Thymoglobulin preparation which is currently applied in clinical immunotherapy.

This paper is excellently written and presented, and includes a good number of well designed and informative color figures.

These experimental observations are mainly of interest for  clinicians. The authors offer a pertinent discussion on the meaning and significance of the observations they have made and described.

Minor comments and suggestions

1.      From an immunological point of view, as a background antibody preparation, a pool of normal rabbit polyclonal sera, conjugated with AlexaFluor-647, should have been prepared, used and tested in a similar way as the specific Thimoglobulin preparation.

2.      References should comply with the instructions given on this matter: For documents co-authored by a large number of persons (more than 10 authors), you can cite the first ten authors, then add a semicolon and add ‘et al.’ at the end: Author 1; Author 2; Author 3; Author 4; Author 5; Author 6; Author 7; Author 8; Author 9; Author 10; et al.

3.      I suggest to include a Data Availability Statement: providing “details regarding where data supporting reported results can be found, including links to publicly archived datasets analyzed or generated during the study. Please refer to suggested Data Availability Statements in section “MDPI Research Data Policies”. 

Author Response

The “Batch-to-batch variation and patient heterogeneity in Thymoglobulin binding and specificity: one size does not fit all” paper describes a deep flow cytometry and induction of cytokines analysis of human Peripheral Blood Mononuclear Cells (PBMCs)  from five healthy donors and using four different batches of a Thymoglobulin preparation which is currently applied in clinical immunotherapy. This paper is excellently written and presented, and includes a good number of well designed and informative color figures. These experimental observations are mainly of interest for clinicians. The authors offer a pertinent discussion on the meaning and significance of the observations they have made and described.

>> Thank you for your kind appreciation and constructive comments.

Minor comments and suggestions

  1. From an immunological point of view, as a background antibody preparation, a pool of normal rabbit polyclonal sera, conjugated with AlexaFluor-647, should have been prepared, used and testedin a similar way as the specific Thymoglobulin preparation.

>> Thank you for this fair comment. We agree that it could have been a useful additional measure to reference, in particular if we could get access to the same rabbit strain that was used to generate Thymoglobulin. In terms of non-specific binding, we observed a large population of cells in whole blood that was not stained by Thymoglobulin, which led us to conclude that binding is due to specific recognition rather than non-specific sticking. Despite not having included this measure in our experimental design (which cannot be fixed on the present dataset, unfortunately), we exclude Fc-R as background issue, as Thymoglobulin bound to all PBMC subsets regardless of Fc-R expression. Appreciation the reviewer’s valid point, we disclosed this limitation:

Page 11, line 337: ‘(of note, we did not have access to serum of non-immunized rabbits to conjugate for reference)’.

  1. References should comply with the instructions given on this matter: For documents co-authored by a large number of persons (more than 10 authors), you can cite the first ten authors, then add a semicolon and add ‘et al.’ at the end: Author 1; Author 2; Author 3; Author 4; Author 5; Author 6; Author 7; Author 8; Author 9; Author 10; et al.

>> Indeed, our referencing complies with this format.

  1. I suggest to include a Data Availability Statement:providing “details regarding where data supporting reported results can be found, including links to publicly archived datasets analyzed or generated during the study. Please refer to suggested Data Availability Statements in section “MDPI Research Data Policies”. Submission Date

>> Agreed. We have now included a statement that allows readers to acquire deidentified raw data upon reasonable request (Page 12, line 376).

Reviewer 2 Report

Comments and Suggestions for Authors

Remarks: Minor Revision

In this article, den Hollander et al. compared different batches of thymoglobulin for antibody composition and variation between individuals in binding to PBMC and T cell subsets, and induction of cytokines. They observed major differences in Thymoglobulin binding to various cell types between individuals, whereas batch differences were minor. Thymoglobulin also induced strong in vitro CRS-related cytokine responses that were again highly variable between individuals but less variable between batches. These inter-individual differences may help to better understand the observed differences in Thymoglobulin treatment efficacy in autoimmune settings. The paper is well written, and the experiments justify the conclusion but formatting the paper for the ease of readers is necessary. The comments are as follows: 

 Major Comments:

 1. The cluster analysis of flow data is very informative, but it is too much information for a reader to process. Could the authors maybe simplify the figure a bit? Maybe add some comparative histogram plots to make the same point or have a bar graph for the cluster analysis?

 2. For Figure 3, the authors mention “Similarly, many shifts were found within the naive T cell cluster, which contains multiple cell surface molecules…”. The authors never mention in the main manuscript or the figure legend what they mean by naïve T cell cluster? The figure only shows CD4 and CD8 cluster. Could the authors specify which cluster they are they talking about when they mention naïve T cell cluster as there is mention of no such cluster in the manuscript or figure or any marker which they use to specify the naïve T cell cluster?

Author Response

In this article, den Hollander et al. compared different batches of thymoglobulin for antibody composition and variation between individuals in binding to PBMC and T cell subsets, and induction of cytokines. They observed major differences in Thymoglobulin binding to various cell types between individuals, whereas batch differences were minor. Thymoglobulin also induced strong in vitro CRS-related cytokine responses that were again highly variable between individuals but less variable between batches. These inter-individual differences may help to better understand the observed differences in Thymoglobulin treatment efficacy in autoimmune settings. The paper is well written, and the experiments justify the conclusion but formatting the paper for the ease of readers is necessary. The comments are as follows:

>> Thank you for you kind appreciation and constructive comments.

Major Comments:

  1. The cluster analysis of flow data is very informative, but it is too much information for a reader to process. Could the authors maybe simplify the figure a bit? Maybe add some comparative histogram plots to make the same point or have a bar graph for the cluster analysis?

>> Thank you for your suggestion. We understand and agree that the cluster analyses and figures can be quite complex and challenging for the reader. We have carefully explored how to best present these complex data for several months. Given the multidimensional character of the results, we believe that we will lose important information if we simplify the figure. Histograms could indeed provide some additional level of comparison for selected parameters but the differences that we noted are often a composite of several markers. More importantly, given the very diverse composition of Thymoglobulin, we refrained from focusing on specific markers but rather a comprehensive view, for which clustering analysis is the way to go.

  1. For Figure 3, the authors mention “Similarly, many shifts were found within the naive T cell cluster, which contains multiple cell surface molecules…”. The authors never mention in the main manuscript or the figure legend what they mean by naïve T cell cluster? The figure only shows CD4 and CD8 cluster. Could the authors specify which cluster they are they talking about when they mention naïve T cell cluster as there is mention of no such cluster in the manuscript or figure or any marker which they use to specify the naïve T cell cluster?

>> We agree with this fair point. We realize that we did not specify our definition of naive and therefore added the specific markers to the manuscript that we used to distinguish the naive T cell cluster (Page 7, line number 196):

“Similarly, many shifts were found within the naive (CD45RA+/CCR7+) T cell cluster, which contains multiple cell surface molecules…”